# Are Bethesda III Thyroid Nodules More Aggressive than Bethesda IV Thyroid Nodules When Found to Be Malignant?

**DOI:** 10.3390/cancers12092563

**Published:** 2020-09-09

**Authors:** Sena Turkdogan, Marc Pusztaszeri, Veronique-Isabelle Forest, Michael P. Hier, Richard J. Payne

**Affiliations:** 1Department of Otolaryngology Head and Neck Surgery, Sir Mortimer B. Davis-Jewish General Hospital, McGill University, Montreal, QC H3T 1E2, Canada; sena.turkdogan@mail.mcgill.ca (S.T.); veronique-isabe.forest@mcgill.ca (V.-I.F.); mhier@jgh.mcgill.ca (M.P.H.); 2Department of Pathology, Sir Mortimer B. Davis-Jewish General Hospital, McGill University, Montreal, QC H3T 1E2, Canada; marc.pusztaszeri@mcgill.ca

**Keywords:** thyroid cancer, thyroid nodule, Bethesda, molecular testing

## Abstract

**Simple Summary:**

The Bethesda classification system is a widespread tool used in the initial screening test for thyroid nodules. The system classifies the biopsy of the nodule into 6 categories, each with its associated malignancy risk and recommendations for management. Nodules classified as Bethesda III and IV are considered intermediate risk, and although Bethesda III nodules are more likely to be benign than Bethesda IV, our hypothesis is that out of those that are malignant, a subset may be more aggressive given their diverse cellular features. In this study we looked at 628 individuals who underwent surgery with a Bethesda III or IV nodule and compared the number of aggressive features found in those with confirmed malignancy. We discovered that Bethesda III nodules that were found to be malignant were more likely to have aggressive features, such as aggressive sub-types of thyroid cancer, spread of cancer beyond the thyroid capsule, and spread of cancer to the lymph nodes. Our results suggest that Bethesda III thyroid nodules may not as indolent as they seem, and these findings may affect management decisions in individuals with indeterminate thyroid nodules.

**Abstract:**

The Bethesda classification system for thyroid fine needle aspirate (FNA) is used to predict the risk of malignancy and to guide the management of thyroid nodules. We postulated that thyroid malignancies characterized as Bethesda III on FNA have more aggressive features than those classified as Bethesda IV. A retrospective chart review was performed to identify those who underwent thyroid surgery at a single tertiary hospital setting between 2015 and 2020. Associations between Bethesda category, molecular genetic test results, and histopathologic findings were examined. Out of 628 surgeries that were performed, 199 (54.2%) Bethesda III nodules and 216 (82.8%) Bethesda IV nodules were malignant. Of those that were malignant, 37 (18.6%) and 22 (10.2%) Bethesda III and Bethesda IV nodules showed aggressive features, respectively (*p* value = 0.014). There was a proportionally increased number of aggressive features in extra-thyroidal extension, lymph nodes metastasis, and all aggressive subtypes of papillary thyroid cancer in the Bethesda III category. Although Bethesda IV nodules are much more likely to be malignant (*p* value = 0.002), our study suggests that Bethesda III nodules that are resected are more likely to have aggressive features than Bethesda IV nodules, with a statistically significant increase in the solid variant of papillary thyroid cancer and lymph node metastasis.

## 1. Introduction

Thyroid cancer is the most common endocrine malignancy, with a nearly three-fold increase since 1975. In 2009, the incidence of thyroid cancer in the United States had increased from 4.9 to 14.3 per 100,000 [1]. Although these rising numbers alongside stable mortality rates suggest an epidemic of overdiagnosis rather than an epidemic of disease, these incidental nodules lead to a greater amount of workup required by endocrine surgeons [2,3]. After the initial imaging, including ultrasound, is performed, the next step in assessing the risk of malignancy of a thyroid nodule is to perform a fine needle aspiration (FNA) [4,5]. According to the American Thyroid Association 2015 guidelines, the cytopathology results of the FNA should be reported with one of six categories as defined by the Bethesda system: I (non-diagnostic), II (benign), III (atypia of undetermined significance or follicular lesion of undetermined significance), IV (follicular neoplasm or suspicious for a follicular neoplasm), V (suspicious for malignancy), and VI (malignant) [4,6,7].

The risk of malignancy for each Bethesda category has been established in the literature and has been updated in the latest 2017 version of the Bethesda system, taking into account non-invasive follicular thyroid neoplasm with papillary-like nuclear features (NIFTP), which is no longer considered a malignancy but still requires surgery for diagnostic and therapeutic purposes. In addition to NIFTP (i.e., borderline tumors), other limitations exist in accurate risk stratification. The malignancy rates of those labeled as atypia of undetermined significance (AUS) are difficult to ascertain, as only a minority of these cases undergo surgical resection. Those that are resected represent a subset of patients with worrisome clinical, sonographic, or molecular findings [8]. Despite this limitation, the implied risk of malignancy (including NIFTP) for a Bethesda III thyroid nodule has been estimated at 10–30%, while that of a Bethesda IV nodule is 25–40% [7].

Pathologists use different cytomorphologic features in the FNA in order to classify them into these categories. For Bethesda IV, the hallmark finding is a cellular specimen with disturbed cytoarchitecture, where follicular cells are arranged predominantly in microfollicular or trabecular architecture with scant or absent colloid [6,7]. In comparison, the list of features to qualify as a Bethesda III category is large and heterogenous. The long list of features implies that there are some irregularities within the cells that are sufficient to be classified as benign (Bethesda II) but insufficient to be classified into Bethesda IV, V, or VI categories. In addition, Bethesda III specimens are often compromised by scant cellularity and/or morphological artifacts (e.g., excessive blood or air-drying artifacts). Importantly, several studies have shown that the risk of malignancy differs according to the nature of the atypia prompting the Bethesda III interpretation [9,10,11,12,13,14]. Specifically, Bethesda III with cytologic atypia raising concern for papillary carcinoma has a higher risk of malignancy (including NIFTP) than Bethesda III associated with architectural atypia alone or Hürthle cells, with a mean risk of malignancy of 47% for Bethesda III with cytologic atypia and only 5% for Bethesda III due to Hürthle cell atypia [9,10,11,12,13,14].

Although Bethesda III nodules are more likely to be benign than Bethesda IV, our hypothesis is that out of those that are malignant, a subset may be more aggressive given their heterogeneous cytomorphologic features.

## 2. Materials and Methods

This is a retrospective chart review of 628 total and hemi-thyroidectomies performed at a McGill University Teaching Hospital in Montreal, Quebec. Data was collected on patient characteristics; tumor characteristics, including pre-operative workup with imaging and FNA; results of molecular genetic testing; and post-operative pathology. The specific type of malignancy (papillary thyroid carcinoma, follicular carcinoma, Hürthle cell carcinoma, or poorly differentiated thyroid carcinoma) and the specific variant (classical, follicular, oncocytic, diffusing sclerosing, tall cell, columnar cell, solid, or hobnail) was recorded. NIFTP tumors were recorded as well. Aggressive tumors were defined as having at least one of the following features: extrathyroidal extension (ETE), positive lymph nodes (LN+), poorly differentiated carcinoma, or any of the following variants of papillary thyroid carcinoma (tall cell, columnar cell, hobnail, diffuse sclerosing, or solid/trabecular) as per the post-operative pathology report. This study was approved by the Research Ethics Committee at the Jewish General Hospital, Montreal, Quebec, Canada. We obtained approval code #37-2020-5791 on November 1, 2019 from the McGill University Health Center REB.

### 2.1. Patient Selection

Data was collected on all patients who underwent surgery with a pre-operative Bethesda III or Bethesda IV classification on FNA between January 2015 and March 2020. Patients who underwent pre-operative molecular genetic testing using ThyGenX^®^, Thyroseq V3, or Afirma were included in the study. Patients with benign tumors on final pathology were excluded.

### 2.2. Statistical Analysis

Associations between Bethesda category, molecular genetic test results, and histopathologic findings were examined. As these were categorical variables, Fisher Exact tests were used. Significance was determined as *p*-value < 0.05. Relative risk calculations were performed STATA/IC 15.1 (StataCorp, College Station, TX, USA).

## 3. Results

Surgical logs for all hemithyroidectomy and total thyroidectomy patients were assessed between the months of January 2015 and March 2020. A total of 628 surgeries were performed on individuals with Bethesda III or IV diagnosis on pre-operative FNA (Table 1). Average age was 52.9 (standard deviation 13.7), ranging from 15 to 88 years old. Of those, 517 (82%) were female and 111 (18%) were male.

Out of the 628 patients who underwent surgery, 367 (59%) had a Bethesda III FNA, while 261 (41%) had a Bethesda IV FNA. Out of those patients, 325 patients underwent molecular genetic testing with either Afirma, ThyGenX, or ThyroseqV3. Fifty-nine (16.0%) individuals showed aggressive features on final pathology. Of the 59, 37 (62.7%) had a Bethesda III and 22 (37.3%) had a Bethesda IV FNA.

When looking only at those with confirmed malignancy on final pathology, we noted that 199 (54.2%) Bethesda III nodules and 216 (82.8%) Bethesda IV nodules were malignant. Of those that were malignant, 37 (18.6%) and 22 (10.2%) Bethesda III and Bethesda IV nodules showed aggressive features, respectively, (*p* value = 0.014). When analysing the differences in aggressive features between Bethesda III and IV nodules (Table 2), we noticed a proportionally increased number of aggressive features in ETE, LN+, and all aggressive subtypes of papillary thyroid cancer in the Bethesda III category. There was a statistically significant difference in the solid variant of papillary thyroid cancer (*p* value = 0.013), defined by a solid component involving more than 50% of the tumor [15]. We also observed a statistically significant difference in LN+, in which Bethesda III nodules were more likely to metastasize to lymph nodes in the neck (*p* value = 0.036). There was no statistical difference in the other aggressive subtypes of papillary thyroid cancer (tall cell, columnar cell, diffusing sclerosing, or hobnail). Similarly, rates of macroscopic ETE were comparable between the two groups. Of note, there were only six (3.0%) follicular thyroid carcinomas in the Bethesda III group, vs. 18 (8.3%) in the Bethesda IV group (*p* value = 0.02). Furthermore, there were 23 (11.6%) and 11 (4.6%) NIFTP diagnoses on final pathology in the Bethesda III and IV groups, respectively (*p* value = 0.007).

Furthermore, our research shows that undergoing molecular testing is significantly correlated with increased risk of malignancy and increased rate of aggressive features (Table 3). Out of 628 nodules, 335 underwent molecular genetic testing; 234 (69.9%) of these were malignant, compared to 174 (51.9%) of those that did not undergo molecular testing (*p* value < 0.001). Furthermore, of those that underwent molecular testing, 43 (12.8%) had aggressive features, compared to 16 (5.4%) of those that did not (*p* value < 0.001).

## 4. Discussion

Our retrospective review demonstrates that resected Bethesda III thyroid nodules are more likely to contain aggressive features when compared to Bethesda IV nodules in our patient population. FNA and Bethesda classification of thyroid nodules is an essential diagnostic tool that is used to better understand the nature of thyroid nodules pre-operatively and to determine subsequent management [4]. Despite advancements in standardizing thyroid nodules with the Bethesda system, preoperative risk assessment remains imperfect as we are currently unable to tell patients with cytologically indeterminate thyroid nodules the type of cancer and risk of aggressive features their nodule may possess. Much of the literature has focused on the problematic entity of malignancy rates of Bethesda III and Bethesda IV nodules, with numbers fluctuating largely between different institutions and regions [8,16,17,18]. Previous papers have attempted to show that Bethesda V and Bethesda VI tumors are more aggressive than other Bethesda categories [19,20,21]; however, to our knowledge, there have not been any studies comparing aggressive features between Bethesda III and Bethesda IV nodules specifically.

Furthermore, an important field for thyroid research includes identifying predictive markers that can be screened for in FNA specimens for the presence of aggressive features. Some studies have suggested that the Bethesda system itself may be used to help predict aggressive phenotypes [21,22]. In their study, VanderLaan et al. demonstrated that the Bethesda system not only stratifies FNAs for risk of malignancy but can also correlate with cancer subtypes as well as the presence of adverse histopathologic features [21]. However, the conclusion to the study suggests that FNAs with atypia of undetermined significance (Bethesda III) identifies low-risk papillary thyroid cancers, supporting conservative clinical management for some patients with Bethesda III nodules. Our study contradicts these findings and the generalized dogma that Bethesda III nodules that turn out to be malignant are rarely aggressive.

A study by Liu et al. in 2016 also concluded that the Bethesda system may be a useful tool in predicting cancer type, variance, and risk of recurrence [22]. Although they noted that nodules defined as atypia of unknown significance, suspicious for malignancy, and malignant (Bethesda III, V and VI) were progressively associated with high-risk disease, LN+, ETE, and margin positivity, nodules categorized as follicular neoplasms (Bethesda IV) imparted unique attributes. Their results noted significantly increased risk of follicular thyroid carcinoma and poorly differentiated carcinoma in the Bethesda IV category, which is in-keeping with trends noted in our study. However, they had greater difficulty in placing Bethesda IV in a continuum between nodules defined as atypia of unknown significance (Bethesda III) and those suspicious for malignancy (Bethesda V), when comparing different high-risk features.

One important factor in studying aggressive features is the classification used to define high-risk in these studies. We chose to include the following features as high-risk classification: presence of ETE, LN+, poorly differentiated carcinoma, or any of the following variants of papillary thyroid carcinoma (tall cell, columnar cell, hobnail, diffuse sclerosing, or solid) as per the post-operative pathology report. These high-risk features were chosen according to the World Health Organization Classification of papillary thyroid cancer and the American Thyroid Association 2015 guidelines [4,23]. However, for the solid variant of papillary thyroid carcinoma, we characterized it as tumors with a solid component involving more than 50% of the tumor with the preservation of the classical cytological features of papillary thyroid carcinoma, as described in previous studies [15,24]. This is also in-keeping with novel studies suggesting tumors with a minor solid component may not be as aggressive [15]. It is also important to note that the solid variant of papillary thyroid cancer is often difficult to diagnose on cytology, which may explain the increased prevalence in our Bethesda III group [25]. Lastly, we also chose to include only macroscopic ETE as an aggressive feature in our study, as microscopic ETE is more subjective and has been recently shown to have no significant impact on outcomes and recurrence-free survival and is therefore no longer a criterion for T3a staging [26,27].

Finally, our study has also demonstrated that undergoing molecular genetic testing is significantly correlated with increased risk of malignancy and aggressive features. Recent studies have shown that molecular testing may be a valuable pre-operative tool to help guide the decision as to the extent of surgery, as certain mutations were found to be associated with more aggressive cancers [28,29,30,31]. Our study highlights the importance of combining Bethesda categories with molecular testing results in order to guide surgical management in indeterminate thyroid nodules.

This population-based study evaluating aggressive features in thyroid cancer by Bethesda grading may have several potential limitations. First, it includes all inherent limitations of a retrospective chart review. As a single center study in Montreal, Canada, a geographic selection bias was introduced. When looking at the outcomes of molecular testing, a selection bias may have been introduced as molecular testing is not currently available at most Canadian centers. It is also important to note that the test is not currently covered by our publicly funded provincial system, and thus the decision to perform the test was made alongside patient preference. All enrolled patients with a Bethesda III/IV thyroid nodule were offered the option to undergo molecular genetic testing to aid surgical decision making. Furthermore, as our institution is that of a tertiary center, the referral pattern may also lean towards a biased increase in patients with aggressive disease. Finally, Bethesda III is a heterogeneous category and the Bethesda III cases in our study were not systematically subclassified (i.e., cytologic atypia vs. architectural atypia vs. Hürthle cell atypia) during the time frame of this study. This may have potentially resulted in different outcomes in terms of risk of malignancy (as shown in previous studies) and type of malignancies for these subgroups. Additional studies should be performed to assess not only the risk of malignancy but also the types of malignancies and their aggressiveness in different subgroups of Bethesda III nodules. Despite these limitations, the findings of our study demonstrate that Bethesda III nodules may not be as innocuous as previously considered and may provide some new clinical applications in individuals with indeterminate thyroid nodules.

## 5. Conclusions

Although individuals with Bethesda IV nodules selected for surgery are much more likely to be malignant, Bethesda III nodules that are resected are more likely to have aggressive features than Bethesda IV nodules, with a statistically significant increase in the solid variant of papillary thyroid cancer and lymph node metastasis. Our study contradicts the generalized dogma that Bethesda III nodules that turn out to be malignant are rarely aggressive. Choosing the optimal extent of surgery in patients with indeterminate thyroid nodules will lead to improved patient care and save the health care system valuable resources.

## Figures and Tables

**Table 1 cancers-12-02563-t001:** Baseline characteristics.

Population	Bethesda III (%) *N* = 367	Bethesda IV (%) *N* = 261
Mean Age		
• 15–30 • 31–50 • 51–70 • 71+	21 (5.7) 121 (33.0) 186 (50.7) 39 (10.6)	21 (8.0) 94 (36.0) 118 (45.2) 28 (10.7)
Gender		
• Male • Female	69 (18.8) 298 (81.2)	42 (16.1) 219 (83.9)
Nodule Size		
• <1 cm • 1–2 cm • 2–3 cm • 3 cm+	17 (4.6) 117 (32.0) 88 (24.0) 144 (39.3)	5 (1.9) 92 (35.2) 62 (23.8) 102 (39.1)
Pathology		
• Benign • Malignant	168 (45.8) 199 (54.2)	45 (17.2) 216 (82.8)
Aggressive Features		
• Not Present • Present	330 (89.9) 37 (10.1)	239 (91.6) 22 (8.4)

**Table 2 cancers-12-02563-t002:** Aggressive features by Bethesda category.

Aggressive Features	Bethesda III 199 Malignant	Bethesda IV 216 Malignant	*p* Value
Tall cell variant of PTC	6 (3.0)	4 (1.9)	0.34
Solid variant of PTC	7 (3.5)	1 (0.5)	0.013
Columnar cell variant of PTC	0 (0)	1 (0.5)	0.36
Diffusing sclerosing variant of PTC	0 (0)	0 (0)	-
Hobnail variant of PTC	3 (1.5)	1 (0.5)	0.21
Poorly differentiated thyroid carcinoma	2 (1.0)	4 (1.0)	0.59
ETE	2 (1.0)	1 (0.5)	0.43
LN+	17 (8.5)	10 (4.6)	0.036

**Table 3 cancers-12-02563-t003:** Comparing malignancy rates and risk of aggressive features between nodules that did and did not undergo molecular genetic testing.

Features	No Molecular	Molecular	*p* Value
Malignant			
- Bethesda III - Bethesda IV	94 (47.2) 83 (38.4)	105 (52.7) 133 (61.5)	0.001 0.092
Aggressive features			
- Bethesda III - Bethesda IV	12 (32.4) 4 (18.2)	25 (67.6) 18 (81.8)	0.011 0.004

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
