# Peer review of "Are Bethesda III Thyroid Nodules More Aggressive than Bethesda IV Thyroid Nodules When Found to Be Malignant?"

_cancers, 2020, doi:10.3390/cancers12092563_

Round 1

Reviewer 1 Report

This is a very interesting paper assessing, whether Bethesda III nodules, when malignant are more aggressive than Bethesda IV nodules.

The methodology is appropriate, the English (as expected) excellent. A high number of nodules was evaluated.

I think the paper is of interest to the readers, however great caution has to be taken in Terms of the Interpretation of the data. As mentioned in the discussion, there is for sure a selection/Center bias in this retrospective analysis.

The conclusion can only be that Bethesda III nodules, WHICH ARE RESECTED... show more aggressive features… (Phrase like this in the conclusion).

In table 1, you mention aggressive features at baseline - what are those? They are not meantioned in the text as far as I see. I have the suspicion that Bethesda III nodules had more aggressive features at baseline, like irregular margins on US, suspicious LN on US, central hyperperfusion, stiffness, high thyroglobulin levels. Because of those, together with Bethesda III classification, the decision for surgery was made. I would like to advise to include those data, as far as it is available, into the paper and also go into more detail in the discussion.

Author Response

Thank you very much for your review and comments.

We agree that this study certainly comes with limitations including a possible selection bias due to our tertiary centre setting.

For the conclusion, we agree with the change in phrasing and this has been adjusted appropriately in the manuscript, including the abstract and conclusion.

As for the aggressive features noted in Table 1 - these were not meant to display aggressive features on baseline assessments but simply the aggressive features on post-operative pathology. This is mentioned in the report discussion. "We chose to include the following features as high-risk classification: presence of ETE, LN+, poorly differentiated carcinoma or any of the following variants of papillary thyroid carcinoma (tall cell, columnar cell, hobnail, diffuse sclerosing and solid)". Our Table 1 is attempting to show that although age, gender and nodule size is comparable between Bethesda III and Bethesda IV nodules, presence of malignancy and presence of aggressive features were not. Unfortunately, specifics of the ultrasound report and lab values were not collected for the purposes of the study. The decision to operate on a Bethesda III nodule is generally made alongside molecular genetic testing (if available), size of the nodule and rate of growth, and patient preference.

Reviewer 2 Report

This is a neatly conducted retrospective study on Bethesda III & IV thyroid nodules with a balanced presentation of methods, results, and conclusions. I would like to propose reorganizing the discussion section according to an earlier BMJ editorial by Docherty and Smith (1999): The case for structuring the discussion of scientific papers. BMJ. 1999;318(7193):1224-1225. doi:10.1136/bmj.318.7193.1224. This would help compressing the section, for instance, with respect to the first subsection “Statement of principal findings”.

The statement “undergoing molecular testing is significantly correlated to increased risk of malignancy and increased rate of aggressive features” (line 121-122) may, indeed, be explained with a potential selection bias (line 186-187) if more serious cases are referred to the best equipped hospital. Please indicate under which circumstances molecular testing actually was applied: did all enrolled patients of a center experience molecular testing if it was available? Or was molecular testing only applied in more serious / more suspicious cases?

Minor corrections:

Line 89: Please delete “a” in “a Fisher’s Exact tests”.

L.90-91: Replace “STATA IC version 15.1” by “with STATA/IC 15.1 (StataCorp, College Station, Texas 77845 USA).”

L.98: Replace “367.” by “Three-hundred-and-sixty-seven” as sentences should not start by numbers.

L.100: Replace “59” by “Fifty-nine”

Author Response

Thank you very much for your review and comments.

We appreciate the advice on restructuring the discussion according to Docherty and Smith recommendations, which has been adjusted with addition of statement of principal findings. We ensured that our discussion generally follows their recommended organization (choosing to leave our paper's weaknesses at the end as opposed to the beginning of the discussion). We opted not to remove much of the discussion, which we think is transparent, fairly compares itself to similar studies, and remains pertinent for readers to make their own conclusions.

We agree that this study certainly comes with limitations including a possible selection bias due to our tertiary centre setting. Because our government does not cover the costs of molecular testing, the testing was done in select situations in which the patient opted to cover the costs. This his been added to the discussion in the limitations section.

Minor grammatical corrections have been made as suggested, thank you.

Round 2

Reviewer 1 Report

I am now fine with the corrections implemented.